# Perceptions of Fear and Anxiety in Horses as Reported in Interviews with Equine Behaviourists

**DOI:** 10.3390/ani12212904

**Published:** 2022-10-23

**Authors:** Suzanne Rogers, Catherine Bell

**Affiliations:** 1Equine Behaviour and Training Association, Surrey GU8 6AX, UK; 2Human Behaviour Change for Life CIC, Norfolk NR9 4DE, UK

**Keywords:** equine behaviour, horse behaviour, owner perceptions, equine fear

## Abstract

**Simple Summary:**

Previous studies have identified that people are poor at recognising fear and pain in horses, consequently leading to one of the key welfare concerns for horses in the United Kingdom. Given that equine behaviourists are uniquely placed to understand how horse caregivers perceive fear and anxiety in their horses, this study examined the experiences of registered equine behaviourists working with horse caregivers. Semi-structured interviews were conducted with nine participants and analysed using thematic analysis. Three key response themes emerged: caregivers are extremely poor at recognizing fear and anxiety in horses; some clients do recognise behavioural signs indicating fear and/or anxiety but only the overt signs (e.g., rearing, running away) rather than the more subtle signs (e.g., tension in face, subtle avoidance behaviours such as a hesitant gait); and fear and/or anxiety behaviour is often misinterpreted or mislabelled. This study has provided initial insights into the lack of recognition of fear and anxiety of horses by their caregivers in the United Kingdom, together with tried and tested approaches to conversations to change this.

**Abstract:**

One of the key welfare concerns for horses in the United Kingdom is lack of recognition of fear in horses. This study aimed to gain an understanding of how well horse care givers recognise fear and/or anxiety in horses by interviewing equine behaviourists (who interact with large numbers of horse care givers and talk to them about this topic routinely). The experiences of Animal Behaviour and Training Council (ABTC)-registered equine behaviourists working with horse caregivers were examined, including the ability of clients to recognise fear and/or anxiety in horses, how clients respond when discussing fear as the reason for their horse’s behaviour, and what explanations the participants use to explain fear and anxiety. Semi-structured interviews were conducted with nine participants and analysed using thematic analysis before being written up to reflect the discussion points. When asked how well horse caregivers recognise fear and/or anxiety in horses, three key response themes emerged: caregivers are extremely poor at recognizing fear and anxiety in horses; some clients do recognise behavioural signs indicating fear and/or anxiety but only the overt signs (e.g., rearing, running away) rather than the more subtle signs (e.g., tension in face, subtle avoidance behaviours such as a hesitant gait); and fear and/or anxiety behaviour is often misinterpreted or mislabelled. These key themes recurred throughout several other interview questions. This study has provided initial insights into the lack of recognition of fear and anxiety of horses by their caregivers in the United Kingdom as well as tried and tested approaches to conversations to change this. Such synthesis of experience and techniques across the equine behaviour sector, together with the information gained regarding perception of equine caregivers, could be a valuable approach to improve the effectiveness of behaviour consultations and welfare initiatives.

## 1. Introduction

Horses display a predictable repertoire of behavioural indicators when they experience painful, frightening, or stressful situations. These behavioural indicators can be overt and dangerous for the human [1], commonly leading to punitive action on the part of the handler and adding to the horse’s distress [2], but they are also present at a much more subtle level. Expressions of pain are now well-recognised in the ‘pain ethogram’ and concept of the ‘equine pain face’ [3,4,5,6]. In particular, the equine pain face has been researched in the context of castration surgery [4], laminitis [5] and pain induced by tourniquet on the front leg and capsaicin applied topically [6]. The inclusion of behavioural indicators alongside physiological markers is recognised as an important factor in the recognition of pain [7]. A wide range of similar behaviours has been associated with fear and stress responses, such as eye wrinkling, twitching and blinking [8], wide or triangulated eyes, ear position, muscular tension, elimination, avoidance and tail swishing [9,10]. Stabled horses were found to show a hierarchy of behaviours with increasing physiological markers of stress [11]. Therefore, given both the welfare implications for the horses and the safety implications for the handlers, it is crucial that people are able to recognise situations that are mildly stressful for horses so that they can intervene and reduce the stressors before the behaviours escalate.

The need for improved recognition of equine pain and stress was identified as one of the four key welfare concerns facing domestic horses in the United Kingdom [12,13]. Similarly, a Delphi study created an expert ranking of welfare concerns, with fear and stress listed as one of the high priority issues [14]. Example triggers of such stress include pain, changes of home and/or ownership and inappropriate riding and ‘use’. Furthermore, it was noted that the ability of professionals working in equine assisted learning to recognise negative affective states has not been studied [15]. Stress during tacking up and mounting was also studied; caregivers were asked whether their horses demonstrated abnormal behaviours when being tacked up, and the horses were subsequently assessed by a vet. Although there was good agreement between the vet and owners regarding biting during saddling and girthing, agreement was less good for evading tightening of the noseband, raising the head and grinding the teeth [16].

In the authors’ original study [17], we asked survey participants to identify equine body language associated with fear and/or stress in a series of videos. Although most respondents correctly identified some of the featured body language, it was common for behaviour associated with negative affective states to be misinterpreted; this was particularly the case for the videos featuring ‘natural horsemanship’ and bridle-less riding and it seemed that the experience and self-identification as ‘equine professional’ of the respondent did not affect the likelihood of correctly identifying the behaviour. There was also a small minority who recognised that the horse was feeling stressed but considered that it was reasonable to continue in that situation [17]. 

Lack of agreement amongst experts regarding body language associated with stress was also found in a Delphi study looking at horses undergoing veterinary care [18]. The sample of experts comprised equine veterinarians, equitation scientists, and animal behavioural and welfare scientists and showed poor conformity on both whether the horses were showing signs of stress and how to describe the relevant behavioural indicators. This contrasts with the findings of Bell et al. [17], where the responses of the expert control group of professional equine behaviourists were consistent with each other.

There are many barriers to learning to recognise and interpret the subtle body language involved. There is an individual element to the behaviours, with some horses more overt than others [18,19]. It is also easy to confuse ‘relaxation’ with the demeanor of a horse who is ‘shut down’, depressed and/or approaching a state of learnt helplessness [20,21,22,23]. It is therefore insufficient to rely purely on quantitative study of whether or not particular behavioural indicators are present, and a degree of nuance is required when interpreting equine behaviour. With this in mind, this qualitative study consisted of a series of semi-structured interviews with experienced, qualified equine behaviourists in order to probe more fully into how fear and/or anxiety in horses is perceived by their clients. In addition, some of the interpersonal elements around the discussions on this topic between behaviourists and clients were examined. A similar methodology has been used to provide insight into the advantages and limitations of the range of approaches used by non-governmental organisations (NGOs) aiming to improve the welfare of working equines from the perspective of NGO staff [24]. 

The objective of this study was to gain an understanding of how well horse care givers recognise fear and/or anxiety in horses by interviewing equine behaviourists (who interact with large numbers of horse care givers and talk to them about this topic routinely). The hypothesis that behaviourists were likely to report that many horse caregivers are unable to correctly identify fear and/or anxiety behaviours in horses was proven to be correct. Despite the variation in clientele, geographical area of practice, and professional background of the behaviourists interviewed, there were key similarities in their perception of the understanding of fear and/or anxious behaviour in horses by their caregivers. Although dedicated support networks exist for professional behaviourists (e.g., support through professional bodies and continuing professional development events), few studies synthesize the experiences of behaviourists in the field. To the authors’ knowledge, this is the first study in the sector to explore the experiences of behaviourists. It is hoped that the collective experiences documented can aid everyone working in the field when considering the discussions with horse caregivers. 

## 2. Materials and Methods

Interviews were conducted through July and August 2022 with a total of nine Animal Behaviour and Training Council (ABTC)-registered, equine behaviourists from across the United Kingdom. Only ABTC registered professionals were invited to be interviewed as this is an unregulated industry and to only include this group removed some confounding factors. For example, a prerequisite for membership of ABTC is that behaviourists use evidence-based approaches to behaviour modification whereas some practicing behaviourists without ABTC membership have not studied learning theory, ethology, or the physiology of stress and use methods that compromise animal welfare. The ABTC registers behaviourists from a variety of courses and routes to qualifications but there are common standards. All individuals were actively practicing although in different contexts (e.g., some saw clients in rescue centres and privately, one predominantly saw race and competition horses). All participants are active as behaviourists seeing a minimum of 35 horses professionally a year and a maximum of an average of 10 each day for those connected with rescue centres in some way.

The researchers initially contacted all 28 people on the list of ABTC-registered equine behaviour consultants (available publicly online). Out of these, nine agreed to be interviewed, eight did not respond, five declined to be interviewed due to the timing of the project, four are no longer actively seeing cases and two are the authors of this study. Through this form of sampling a volunteer sample of participants was recruited. While the small sample size might commonly be regarded as a problem for studies that rely on a statistical analysis of large numbers of participants, in a qualitative study it provides a greater depth of understanding about the participants of the study and their perceptions regarding a larger number of clients; and there were high levels of agreement between participants. It should not be assumed that these results can necessarily be considered representative of the equestrian community more widely but, taken in the context of the authors’ previous quantitative study with a larger sample size [17] it would seem likely that they are indeed fairly representative. Previously, we found that equestrian experience was not a predictor of the ability to recognise small signs of fear and stress but that clicker trainers showed a possible increased ability. Furthermore, considering the number of practicing active behaviourists in the United Kingdom, the sample size is fairly proportionate. This study focusses on what the participants said, based on their extensive experience with clients, rather than how many participants gave certain responses.

Interviews were pre-arranged and took place online using Zoom 5.11.1 (6602) video conferencing software. For subsequent transcription, all interviews were recorded, and ranged in length from 9 min 57 s (Participant 5) to 37 min 51 s (Participant 3). Participants received a document detailing information about the project and regarding consent before interviews were conducted. Participants gave their verbal consent to the process at the beginning of each interview. 

Participants were asked eight questions on the theme of their experience with clients regarding fear and/or anxiety behaviour in horses. Interviews were semi-structured, so the question script was flexible, and clarification was sometimes requested. Semi-structured interviews were chosen as the qualitative research tool because they provide a flexible technique for small-scale research [25], provide useful data from small sample sizes and allow thematic analysis of the qualitative data [26].

The interviews were transcribed using an automated online service and edited to remove duplicate words, and the phrases ‘I know’, ‘You know’ and ‘Um’. The grammar was edited in places; for example, adding ‘full stops’ because often the automatic transcription created very long sentences. Participants were assigned codes P1 to P9. A thematic analysis was conducted [27] for each question to identify themes that ran through all responses and the answers written up as a narrative. Data analysis followed Braun and Clarke’s six-stage process of thematic analysis [27]. Familiarisation (stage 1) involved reading the transcripts to remind the researcher of the interview content. Coding (stage 2) was performed on the transcripts from the nine interviews to aid the generation of key themes. Coding was done manually using the highlighter function in MS WORD (Microsoft® Word for Microsoft 365 MSO (Version 2209 Build 16.0.15629.20200), to colour similar phrases, concepts or patterns that were repeated throughout the interviews [28]. These were then printed out, cut up (i.e., each coloured section was cut ‘out of’ the interview taking care to label the pieces with the participant code number) and ‘clustered’ (putting similar content together) to enable the generation of themes (stage 3) representing the repeated concepts or elements. These draft themes were reviewed (stage 4) as sometimes emerging themes were combined into one main theme or separated into two separate themes. The themes were then defined and named (stage 5) and finally written up as research results (stage 6). Next the transcripts were all read again in their entireties to extract full quotes that were relevant to the emerging narrative. Finally, the transcripts were ‘cut up’ so that each participant’s answer to each question could be compared, summarised and discussed alongside each other.

## 3. Results

This section outlines the results of the thematic analysis that was used to examine the responses to each question in turn. A summary of the questions and themes for each one is given in Table 1. 

When asked, as a warm-up question, what prevalence of their caseload has some element of fear and/or anxiety, the lowest percentage given was 80%. Responses included *“I would say probably all of them”* (P5), “*9 out of 10 cases*” (P7), and “*I would say 95% of the cases involve some form of fear or anxiety*” (P9).

### 3.1. Question 1: How Well Do People Recognise Fear and/or Anxiety in Horses?

When asked how well horse caregivers recognise fear or anxiety in horses, three key response themes emerged: caregivers are extremely poor at recognizing fear and anxiety in horses; some caregivers do recognise behavioural signs indicating fear and/or anxiety but only the big signs (e.g., rearing, running away) rather than the small signs (e.g., tension in face, subtle avoidance behaviours such as a hesitant gait); and fear and/or anxiety behaviour is often misinterpreted or mislabelled. These key themes recurred throughout several other interview questions too, to become the themes of the whole study. All participants covered all three response themes in their answer to this first question to some extent. 

Regarding the first and second themes, answers to this question unanimously focussed on how owners do not recognise the behavioural signs of fear and anxiety well at all and especially not the more subtle signs. For example, “*It is very rare that anyone actually recognizes it genuinely. I’d say that in the majority cases I see, often they understand it or recognize it, but only once it’s got to a certain level. So, it’s only once it’s got to a horse going essentially over threshold. […] I can think of only maybe two owners out of hundreds that noticed a fair anxiety at a lower level, in terms of body language, facial expressions, vocalization, that sort of thing. […] It’s just non-existent because it’s not taught.”* P7.

Regarding theme three, mislabelling behaviour (as ‘naughtiness’, for example) was mentioned unanimously, some examples follow (note that often responses included all three themes intertwined): *“They usually will not have recognized those small behavioural signs. […] I think most of them would only accept actual attempts to create physical distance. They might see that as fear, if they didn’t interpret it as be naughty or disobedient, or not wanting to do something.”* P1.*“I think they recognize fear behaviour, but they don’t always label it correctly. […] Often it’s maybe misinterpreted to be naughty, cheeky, excited, being a pain, whatever, and not identified as a fear-based response, or anxiety. Especially with anxiety it’s often seen as like excitement—high level energy.”* P2.*“I think people often mistake a lot of avoidance behaviours for being cheeky or being naughty. And then they end up putting themselves in a situation that becomes dangerous.”* P9.*“I had a great pony this week who was a brilliant example and could not have been showing more avoidance behaviours if he tried. I’d just been told just before seeing him that, ‘Oh, he does this, but he’s not scared’. I mean he pretty much showed every subtle avoidance behaviour he possibly could before he started running around. But they just weren’t recognized at all. […] I would say almost all of the cases that I’ve had where I’ve gone out to an aggression case has never really turned out to be that at all. You get there and it’s just like aggression is not even part of how I would describe what the horse is demonstrating. Usually, it’s just fear signs mainly.”* P3.P4 described how owners often label certain character traits in the animal and fully accept that the animal is just ‘like that’, without considering the reason for the behaviour.

An interesting reflection explored how some clients seem reluctant to label fear behaviours as fear or anxiety but seek explanations reflecting that they recognise their horses are struggling in some way. *“Well sometimes*
*they over interpret it as being some sort of PTSD* [Post Traumatic Stress Disorder]*. It’s almost as if there’s some sort of reluctance to use the terms, fear, or anxiety, and to kind of over egg it as some sort of really serious PTSD relating to that horse and some sort of history.”* P1

Some interviewees mentioned variation across their client base (e.g., owners of cob type horses, which often show less overt fear behaviours, were reported as being less able to recognise fear and anxiety than owners of Arabs or thoroughbreds, breeds that are generally considered to be more reactive). For example, *“It does vary quite widely and there is probably less consideration of it in a competition context or a performance horse context”* P9. This participant also reflected that there is a disparity between different sectors of the equestrian population—that staff in rescue centres usually recognise fear components of horse behaviour whereas competition and racehorse caregivers tend not to even consider fear or anxiety as being part of the issue.

One interviewee (P3) directly reflected on the type of client they see describing how the vast majority of their clients are not able to recognise fear and/or anxiety in their horses but that some who are very dedicated to their horses and interested in behaviour are better at recognising the overt signs, however, even these people still miss the more subtle behaviour signs. 

P4 reflected: “The majority of horse clients have some recognition that there is some fear of anxiety, probably 80% of them. The [other] 20%—although it would not be a formal ‘I know my horse is scared’—in some way their language would reveal that they have identified ‘I know he is a bit worried or yeah, perhaps he’s a bit scared’ or those sorts of things. This is a thoughtful reflection, and it would be interesting to do a lexicographical study regarding language used by clients in future research.

### 3.2. Question 2: How Do Caregivers Tend to Respond When Told Their Horses Are Fearful and/or Anxious?

All the participants discussed how their clients have several possible responses that upon thematic analysis fell into two groups: those clients who take the information that their horses are fearful/anxious on board; and those that are sceptical of fear/anxiety being a factor and continue to label the cause of the behaviour as something else (e.g., horse is being lazy or naughty). The latter group fits into the earlier theme of misinterpretation of fear and/or anxiety in horses. The following quotes illustrate these two groups:*“Some will say, ‘Oh, that’s interesting, I didn’t realize that, and that makes sense now you’ve explained it to me’ and others will go ‘Oh, no, he is not frightened, definitely not frightened’ and give me justifications for why they think the horse isn’t frightened.”* P1*“I think sometimes they don’t believe me. Sometimes they just think the horses are still being quite difficult and they can’t quite understand. They just think the horses would always run away if they’re fearful—that they wouldn’t get pushy and bulky, if they’re fearful.”* P5P4 expressed that a lot of people do not realise the degree of fear their animal is experiencing or that the fear is responsible for the behaviour they consider problematic.P3 reflected that, in their opinion, whether or not people appreciate their horse is fearful is linked to their own agenda. So, if they want to ride in the woods, or do competitions, they ‘need’ to feel that the horse enjoys it too and are less likely to accept that the horse is fearful than, for example, people who can empathise with a fear of injections. They conclude *“And then that’s the horse world as a whole, isn’t it really? It’s all about what the people want.”*

Several people mentioned the very strong emotional attachment many of their clients have to the horses they look after and that hearing the horse is fearful is very difficult to accept. 

Participants again gave examples that using thematic analysis illustrated that different types of clients respond differently when they are told their horse is fearful. For example, several participants highlighted that staff in rescue centres have generally, but not always, received appropriate training in recognising fear behaviours and respond positively to the behaviourist’s explanations. Most rescue centre caregivers and leisure horse owners, however, are receptive to some degree and most owners of competition and racehorses are the most resistant. 

One participant explored how sometimes people’s aims and desires regarding what they want to do with the horse ‘blind’ them to the horse’s welfare. *“I think I do have some contexts where people will recognize that those signs are there but actually that doesn’t fit with what they might want to do with the horse. So, for example, I have quite a lot of racehorses that I see that have either kind of loading issues or issues on the stalls or tacking equipment issues. And actually, whilst they want to solve the issues, removing that stressor isn’t a factor for them to consider. […]”* P9.

### 3.3. Question 3: Is Fear/Anxiety Something You Tend to Always Talk about in a Consult or Just If It Is Their Primary Reason for Calling You Out?

When asked if they would usually mention fear/anxiety if they saw it was an issue but perhaps if it was not the main reason they were called out, two themes emerged. First, those interviewees who said ‘always’ (six out of nine people), e.g., *“Generally, I would always try and find a way to weave it in somehow”* (P3) and *“Always fear and anxiety would come into the discussion if the animal was exhibiting any of those signs”* (P4). Second, those who would be more cautious (three out of nine people, e.g., *“It would depend on the owner”* (P7) and *“Only if it’s relevant, which it often is, but […] generally I would keep it contact specific”* (P7).

### 3.4. Question 4: What Explanations/Analogies Have Worked Best in Getting across to People That Their Horses Are Fearful and/or Anxious? 

Three main approaches to explaining that horses are fearful or anxious emerged from the discussions:Likening their horse’s fear/anxiety to human experiences of fear/anxietyFraming explanations from the angle of ethologyFocussing on observations of the horse in front of them.

An example of likening equine fear to human experiences was described by P4 *“I a**lways use human examples of phobias or fears; the common ones being spiders, heights, claustrophobia, those sorts of things.”* and *“I guess through my work I’ve seen people come back and go, ‘You know, when you talked about spiders that really resonated’, or ‘When you talked about heights, I really got it. I didn’t think about it in that way.’ […] I’m not sure the first element is the visual element for me. I think it’s trying to get people to feel the fear and anxiety so that they can then use their mirror neurons to understand what might be going on. I do this by asking, for example ‘Do you ever hide your fear or worries in certain situations? What would you do on a first day at work? Would you confess to being terrified of whatever it is, or would you try and mask that? And why would you do that?’ So again, drawing out from them, the possibilities and then drawing some comparisons between the two.”* This participant continued to explain *“And most people suddenly discover that it’s not spiders per se. It’s the movement of a spider. It’s a big spider, it’s an unpredictable spider. And that’s the same for heights. It’s often not a fear of heights, it’s a fear of falling. It’s a fear of losing control and jumping off to find out what happens. And if you can dig into that a little bit, it allows them to at least appreciate the details. Doesn’t have to be logical for the horse either. It is what it is.”*

The example of first explaining fear using the example of humans who are scared of spiders was also described by P1 and several other interviewees use similar examples and also first ask what the horse owner is scared of and tailor their explanation to that example. 

An example of framing the explanation in terms of ethology was given by P2: *“I generally go back to if the vet was a lion, your horse [..] doesn’t want to show that they’re scared”*. This participant continued to explain how they describe how horses being prey and herd animals affects their behaviour, and hence how they ‘hide’ behaviour so as not to draw attention to their weakness to predators.

One participant, P6, framed the explanation in terms of perception: *“**So for example, a puddle to you might just look like a puddle, but to a horse, that doesn’t really have that same depth perception, they might see that as a bit of a pit.”* P6 continued to describe how asking horses to stand with a dental gag in their mouth, having work done with an electric drill, is “crazy” to expect from the perspective of a wild horse or donkey, concluding “*I think both of those things, highlighting the perceptual differences and highlighting how strange domestication is, is quite effective.”*

The third approach, focussing on observations of the horse in front of them, was mentioned by several interviewees. Examples included:***“****We’ve usually got the horse in front of us and will be pointing out like the tiny little signals that they’re giving and tiny little changes in their face and the really subtle body language movements and stuff like that.”* And *“I talk quite a lot quite early on with people usually about how there is no such thing as a naughty horse [..] And I talk quite a lot about individuals, horses not necessarily fitting into certain boxes—so you just need to look at a different way of doing things rather than asking the same thing.”*P9 *“I quite often use the communication ladder because I just think that’s a really nice, clear explanation.”* This participant continued to explain how describing a ‘ladder of communication’ introduces the concept of escalation in a way that owners respond well to as they recognise some elements, and can develop an increasing awareness of other elements, and recognise that if they ‘keep pushing’ the horse, then he/she is likely to escalate their behaviour.P2 outlined that they explain the more overt signs of fear and/anxiety in horses are effectively the horse ‘shouting’ that they are not being heard and are having to show ‘big’ behaviour. The interviewee continued to explain that if the human caregiver can start to ‘listen’ to what the horse is communicating, and recognise the smaller signs, and respond to them, the safer it is for everyone.P5 *“You start to get people just to watch. You can get people to recognize subtle signs. You just need to give them the chance to observe rather than ‘Let’s get on and do it’. It takes a bit of time and a bit of talking through it.”*

Several participants described tools or approaches that were not mentioned by other participants. For example, using pictures, lists and examples of work done with other clients with fearful horses: *“I have a handout I use quite a lot, which is a very pictorial explanation of trigger stacking and it basically puts it in a human way […].”* P9 describes how they find that it helps owners to understand how they might feel and in turn how their horse might feel. P9 continued to explain the value they have found in using pictures over verbal explanations *“My explanation might be not what they’re imagining, so I find pictures are quite useful.”*


P9 also described how asking owners to make lists is something they have found useful to encourage active participation from the owners. For example, this interviewee asks owners to make lists of what their horse ‘likes’, ‘does not like’ and ‘really does not like’ and categorises them as green, amber, and red, respectively, and then discusses the list with the owner. 

Two participants, P8 and P7 explained how they talk to owners about other cases (anonymised) that they have worked on that are like the client’s case. P7 explained how this approach seems to help owners feel better about their situation and feel like they’re not alone. *“They just generally calm down quite a bit and they become a lot more receptive to any advice you may give because they feel like they’re not alone.”* P8 suggested that this approach works due to the human tendency to respond to being in a peer-group (e.g., group of people experiencing similar issues with their horses) and draw comfort from such social support even if only through hearing the behaviourist talk about clients they have helped with similar concerns. This participant also talked about the benefits of social support in terms of helping clients to seek social support, and/or to signpost them to certain social media groups, for example. P8 also highlighted how providing clients with resources, such as research papers and magazine articles, helps them to feel less ‘alone’.

P7 described another tool—the use of scales to consider spectrums of behaviour, which is similar to the ladder of communication that P9 described. *“I talk about a threshold a lot. I use the scale of one to 10 with people because it’s a really easy way for them to quantify and understand the fear and anxiety. So, we often discuss, if a horse is spooking at something, by the time we actually see the spook, they’re probably at nine or 10. They’re at or above the threshold. And then before that you might see some subtle vocalizations or face expressions […]. And what I often do is we often try to look at that in a scale so we can get people to understand how those sorts of expressions of model language build with the stress and fear that come up so that they can stop, start to notice and anticipate fear occurring and try to modify their behaviour, and how they respond to their horses early on to avoid it going any further.”*

It would be interesting to conduct further analysis regarding the educational props and exercises different people use in behaviour consultations and also to get input across species to pool and build on ideas. It is important to note that although the three themes came out between all participants, they might well use more than one but not have focussed on it in the time constraints of the interview.

### 3.5. Question 5: Do you Come across Resistance? People Not Believing Their Horses Are Fearful and/or Anxious?

All participants explained how they tend to approach conversations where the horse caregiver appears resistant to the explanation that their horse is fearful and/or anxious. A clear theme emerged on encouraging the owners to reflect on their own behaviour and understanding of fear in horses and humans alike. An example that represents this way of thinking was given by P3 *“I think probably what I try to do is turn it around on them a little bit in that situation. And I question them a bit about it. So, ‘He does this, but he’s not frightened’, and so I’ll often turn that round to ‘Okay. That’s really interesting. What do you think is the reason why he’s doing that?’ And if they say it was just being naughty, then I would usually say something like, ‘Well, there’s a reason for every behaviour. So, what do you think the reason would be for that? Why would he be being naughty?’ And that usually quite quickly ends in them not being able to really come up with an answer.”* This participant continued to reflect on how they deal with resistance by being as interactive as possible, for example through using video clips and discussing observations of both the handler and the horse. 

P4 explained how people fear different things, and when faced with something they are not scared of but someone else is, can sometimes try to diminish that person’s fear, and that explaining this might be happening can help people to understand their horses’ fear. 

A second theme that emerged was that the interviewees also made several reflections regarding the interpersonal skills needed to be an effective behaviourist. For example, P4 *“I think human nature needs a little bit of that time for them to process and the mistake that we’ve probably all made in our enthusiasm, especially at the start of careers and things like that is to try and tell people instantly, ‘Can you not see how terrified he is?’ I still think we’re for a very short on human behaviour skills when we’re trying to convey these messages to people and allowing them to work out for themselves.”* And *“I think it depends on the skill of the practitioner*.” P2 observed *“You have to be really careful not to trigger that defensiveness just to let them explore it in their own sort of way. It’s a real tight rope sometimes.”*. P7 highlighted the importance of good communications skills *“You have to be very aware of the language that you use and the way that you approach [people]. But I find that a lot of people do try to sort of change the narrative—they try to make it seem less bad than it actually is.”*

P1 reflected on a slightly different element of the professional working element of dealing with resistance *“**I […] rely on my expert status […] I don’t try again and again, to persuade them. I’ll explain it to them once or twice and if they don’t accept it, then, […] I’ll still include it in my report, and I will base my treatment plan and behaviour modification plans on that and send them information and reading material. And that might kind of bring them round to an understanding, but sometimes, you can’t. You can lead a horse to water, but you can’t make it drink, and you can lead a client, you can give the client the information, but they might not be in a place where they’re ready to accept it. Because of their values, beliefs, their own learning history, what owning a horse and riding a horse means to them socially, personally, and psychologically.”*

Again, some differences between types of clients were mentioned. P9 stated they come across more resistance to the idea that behaviour is caused by fear/anxiety in the horse racing sector than others. Also, *“Then you get the opposite end of the spectrum with lot of rescue places that are just so switched on and taking everything really slowly and doing all the right things and that’s great.”* P5 also reflects on resistance in rescue centre staff—*”If they’re very traditional, ‘Horses just need a good smack and tell what to do’. They’re the ones that are quite difficult to persuade any different. I suppose, some of its fear based from their point of view as well. Cause some of the rescue horses are quite wild. You’re not dealing with your normal everyday horse quite often.”* Again, also there were reflections about the type of clients that tend to approach each behaviourist *“If they’re very traditional mindset, horse people, I generally don’t see them for a start.”* P2.

### 3.6. Question 6: Do You Find There Is a Difference in How People Perceive Fear and/or Anxiety If the Cause of It Is a Thing (e.g., Hose, Car) or If the Cause Is the Owner Themselves (e.g., Training Method)?

When answering this question all participants described case studies and reflections comparing how owners perceive fear when horses fear objects (e.g., bins, parked tractors) and when horses fear the owner’s behaviour. The key theme through all responses were that people are more compassionate to the horse when they are fearful of an object than the behaviour of a person themselves. Reasons focussed on the fact that being afraid of ‘something’ is easier for people to understand, and that the relationship between horse and human, from the human’s point of view, is often a complex mix of emotional attachment and utilitarian philosophy. People want to ‘use’ horses for their leisure activities, but it is important to them to also have a relationship with that animal and for the animal to ‘love them’. The behaviourists all brought into question whether the emotional attachment is so positive from a horse’s perspective. For example: P6 ***“****I think people are really open to the fact that horses can be scared of going in the trailer or scared of the vet or scared of these husbandry procedures. I think people don’t want to accept that their horse might fear them.”*

P4 also noted that sometimes people just cannot comprehend why a horse would be fearful of something that they, as a human, are not. Using the example of a rubbish bin, P4 noted that to encourage someone to appreciate that their horse is fearful requires exploring the logic from the horse’s point of view—perhaps they had a bad experience with a bin, perhaps they fear the noise it makes when moving, and so on. 

Conversely, P7 described how clients often cannot understand why their horse fears a car but have more empathy for a horse who is scared of a veterinarian trying to give an injection, as they know people who do not like needles or do not like needles themselves. However, this still fits with the idea that people are better at recognising fear in horses if they can empathise regarding the cause of the fear. 

### 3.7. Question 7: Are Things Changing? 

This question generated a spectrum of responses from positive to negative. Often as interviewees continued to talk, they tended to go from one end of the spectrum to the other, irrespective of which end they started at. Some examples:*“I think it is changing. I think there are groups of people who are much more aware of it than there were 22 years ago. When I started, it was kind of like ‘Scared? Don’t be stupid!’. Now, you think everybody gets it. So, I would say it is changing and there is more recognition in many groups and areas and individuals, but there is still a massive majority that do not recognize the size of the animal’s fear, or the frequency of fear in equines.”* P4*“I’m not sure if people are [changing]. There’s definitely change in the world but I’m not seeing too much. I heard some good things from a coach, who comes and does riding, and instructing at my livery yard. I have heard him talk about positive reinforcement and stuff, but then when I’ve watched lessons, it does go into flooding. I think some individuals are on board, but I think there’s a lot that aren’t. I know my vets actually have on their website professionals they refer to, but none of them are behaviourists. They’re all trainers, there are no behaviourists. Maybe we’re getting there, but I’m not sure yet.”* P6P2 described how they have seen an increase in discussion regarding the social license to operate with reference to equestrian sports and an increase in social pressure to consider equine behaviour in terms of why the horse might be doing something. P2 continued to describe observations that there is perhaps an increasing exposure in the media regarding considering behaviour from a more scientific perspective. For example, they observe on social media more people mentioning the possibility of pain causing some unwanted behaviours without the majority using words suggesting behaviour is being misinterpreted as the horse being ‘naughty’. *“I don’t think it’s changing willingly, and I think there’s still this weird kind of, ‘I love my horse, but he’s got to have a job to do. […] I think that this is stopping the big changes that we would all like to see. I think there is a shift in the right direction, which I think is being fuelled by maybe more education in colleges and work on pain.”*P7 described positive changes they have observed such as more veterinarians referring cases to behaviourists, but also highlighted that the referral rate is still very low, often referring behavioural issues to riding instructors.P1 *“For many, many years I was very despondent and thought there wasn’t any change, but I would say in the last two or three years, really, I have seen more of a change and, but not so much in terms of recognizing fear and anxiety, but in terms of accepting or understanding that it can exist, that it can be a factor in a horse’s behaviour.”*

Some interviewees reflected on approaches that could be taken to facilitate change and challenges in doing so. For example, P7 described the fine balance of trying to engage the racing community to drive positive change without alienating them. *“You want to make huge steps forward in welfare, but equally you want to bring everybody with you while you’re doing it.”* P7 also described how there is an increasing body of research and evidence, but challenges in transferring the science into practice.

Several people questioned whether their perception of change is accurate or biased due to various elements of their working context. For example, P3 said *“I d**efinitely do think it’s changing. I do feel like there’s a massive shift, but I don’t know whether I also feel like I maybe have a slightly skewed view of that because most of the work that I do now is in a welfare organization.”*

The term ‘social licence to change’ was used by several interviewees to reflect that societal acceptance of some elements of equestrianism seems to be changing and several interviewees provided examples of observations from social media interactions changing in time. For example, P3 *“Even not that long ago, three, four years ago, if someone put a question on like, ‘Oh my horse doesn’t want to load’ or something, you’d have 15,000 ways of getting the horse in, in a very unpleasant manner. Whereas now I would say it’s starting to balance out a bit. I think there’s a lot more people that seem able to be the voice for the horse on there these days and there’s always people that are going to shoot them down as well. But I do definitely think there’s been a shift. And especially, I would say in the last one to two years.”*

P5 expressed that recognition of fear and/or anxiety depends on the methods and horsesector celebrities people have been watching or listening to and said *“I used to ride with some very old ladies, and they were actually quite good. Back in the seventies and the eighties, when we used to ride, they seemed to really understand, whereas I think it’s shifted the wrong way, but potentially it’s shifting back.”*

P8 described how the large charities are changing significantly but that in their opinion, mainstream and sport-orientated clients are not changing. *“I**t’s the big protagonists that are changing, some of the bigger charities that I would’ve thought it was almost impossible to change, (especially those with overseas links) are trying, they’re really trying”.* This participant also highlighted the complexities surrounding how clients are more or less likely to take on information depending on the professional giving it. 

### 3.8. Question 8: Is There Any Other Information You’d Like to Share about Your Experiences with Your Client’s Perception and Understanding of Fear and/or Anxiety in Horses?

When asked what other information the participants wanted to share regarding their experiences with horse caregivers’ perception and understanding of fear in horses, several barriers were highlighted. The issue of ‘time’ was highlighted by several participants. For example:P4 *“People struggle to understand the length of time it takes to overcome a fear. My experience is that equines get over fears and phobias much quicker than humans, when it’s done well, because they don’t seem to hang onto it deliberately, humans are justifying their fears. But that’s not quick enough for most people. […] A challenge I see for the behaviourist is to be able to get people to buy into the benefits of two or three months of work, to remove a fear before you even get to the point where we can have a safe vet visit or load on the trailer or walk past the dustbin or whatever it might be. I think that’s because human beings just trade future benefits for current ones all the time.”*P2 *“I think people really love their horses and they really want to do right by them and not have them be scared of anything. However, I think they find it really annoying when they can’t go out and do what they want to do with their expensive animal that takes up lots of their time.”* P2 continued to describe how clients seem to have a limit to their patience in terms of the timescale they ‘accept’ for their horses behaviour to be ‘sorted out’ even if they accept their horse is scared or in pain.

Another theme that emerged from this last section of the interviews was the barrier regarding the norming of fear behaviours in equestrian society. P9 *“It is desperately trying to get rid of that naughty label, the biggest challenge, because it’s just been sort of generations, hasn’t it? Generations of being cemented into society’s perceptions.”* And *“I find that people have very often not even considered gradually introducing things to the horses—it’s almost a given that the horse should be able to stand and be shod because why wouldn’t it? As opposed to going, ‘Okay, this is a really bizarre process, and it needs to be introduced really gradually.’”* P3 added: *“I wonder whether a lot of it comes from a kind of social pressure from the industry as a whole, that you can’t be seen to be fluffy—because then people will think less of you or something. I think there’s a real stigma in the horse world around people who actually treat their horses like proper individuals and sentient beings. I guess when you start thinking about it, it is quite scary because I think essentially the equestrian world is just, what’s the word, sort of a culture of violence, I suppose. There is talk to kids from a really young age that you can make that animal do whatever you want by whacking it by buying your child or sparkly whip when they’re five years old. And that breaks my heart. I just think violence is so normalized in the horse world. Isn’t it? There’s such cognitive dissonance. I don’t think people can accept often that their horse is showing signs of fear or anxiety because then they have to question everything they do and everything they’ve done.”*

P6 reflected on the prevalence of horses showing pain, fear or anxiety in the public arena *“**If you watch dressage or show jumping on the television, or at a show, you see so many pain faces and fear faces that people just don’t recognize. Maybe because it’s so normal for horses in those contexts—they just don’t look very happy.”*

Several participants mentioned cognitive dissonance. For example, P6 *“People do love their horses and they do want to do right by them. So, while they can accept certain things might be scary, they can’t always accept that they are the reason their horse is scared. Not without some cognitive dissonance.”* P3 also described the disconnect that cognitive dissonance creates; describing a case where ponies were showing quite a lot of subtle avoidance behaviours (e.g., turning away, facial tension, triangulated eyes, hesitant gait and so on) and yet because the pony has not moved the handler interprets their behaviour as ‘fine’.

P7 described the advertising and marketing associated with natural horsemanship approaches. *“**They do create a lot of harm in terms of the perception of fear because they obviously are using a lot of aversives and they’re creating fear through training. But they obviously talk about it in a way that makes it so magical and sounds like so mysterious and so lovely.”* And *“Then when you start to change that perception, or we start to question that perception in front of them, and it’s almost like opening the matrix a little bit, isn’t it? I think that’s how someone explained it to me. They said you’ve turned up and you’ve literally turned my world upside down and it’s like, you’ve literally, lifted the matrix and I can see everything again clearly.”* The interviewee concluded that *“I guess that’s a nice response to have, but equally, what does it say about the equine industry itself?”.* This participant also reflected the link between improved horse welfare and increased safety. *“But I don’t think people make that link. I think a lot of the time when people think about safety, they’re thinking about controlling the horse, not necessarily making the horse any more comfortable. I see that a lot where people are, they’re looking just for the next gadget or the next training method, that’s going to make their horse do whatever they want to do at whatever time they want them to do it. Because that’s how they feel like they’re in control and they’re safe.”*

## 4. Discussion

The results of the thematic analysis, together with the summaries of the responses to each interview question, provide an insight into the perceptions of fear and/or anxiety in horses by their caregivers as perceived by equine behaviourists. Throughout the interviews, responses from the different behaviourists were consistent and even the case studies they described shared key elements. This consistency within behaviourists is an important point because it shows that the reason that there is an inability to identify fear and anxiety behaviours in horses is not that horses are inconsistent in their fear responses, or that their behaviour is open to varied interpretation (as could be assumed given the lack of recognition of fear behaviours). Although horses are prey animals and typically display subtle signs of pain, fear and stress [29], the consistency with which behaviourists recognise these subtle signs suggests that the subject can and should be a key part of equestrian education. Such consistency contrasts with the findings of Pearson et al. [18], that different professionals disagreed on recognition of stress and how to describe its behavioural indicators, perhaps suggesting that the education of behaviourists includes key information that should be extended to other professions within the equine industry.

This work builds on previous work [11,13,17] that identified an inability to correctly interpret ‘stress behaviours’ in horses. This affects horse welfare because if caregivers do not identify when horses are fearful, anxious or ‘stressed’ they are likely to continue interacting, handling and training horses in a way that causes those negative affective states, and consequently cause behaviours that could be dangerous, as well as compromise horse welfare. The hypothesis that behaviourists were likely to report that many horse caregivers are unable to correctly identify fear and/or anxiety behaviours in horses was proven to be correct. 

Anthropomorphism influences the perception of animal emotions both negatively and positively [30,31]. This is reflected in the participants’ responses not only regarding anthropomorphism used by their clients but also by the participants themselves. Behaviourists are skilled in using ‘constructive anthropomorphism’ [32] to help their clients understand the cause of their horses’ behaviours and to encourage their clients to develop empathic responses. Discarding the potential that horses feel these emotions is likely to reduce such understanding and empathy, to allow the maintenance of client behaviour that causes fear and ultimately to reduce animal welfare.

A limitation of the study was the small number of behaviourists that took part. As explained in the methodology, the authors wanted to limit participation to those equine behaviourists who were registered under the ABTC. However, this significantly limited the pool of people from which to sample to participate. If wider inclusion criteria had been included, a greater number of equine behaviourists could have been interviewed. Given that several interviewees mentioned that different types of horse owners (e.g., owners of leisure horses, racehorses, and competition horses) correspond with differences in perception of fear and anxiety, the study was limited by not more carefully following this line of questioning to gather data that would enable more concrete conclusions to be made regarding differences between the owner types. 

The study highlighted several areas for future research. First, it would be interesting to assess whether the perceptions of other equine professionals (e.g., farriers, equine physiotherapists and riding school instructors) regarding the perceptions of equine fear and/or anxiety in horse caregivers follow the same themes and highlight the same topics as this study has. Second, an investigation into the use of language regarding behaviours associated with fear and/or anxiety in horses would further understanding of some of the complexities of communication on this topic. Third, a study to examine what interpersonal factors affect clients’ likelihood to act on advice given by behaviourists and veterinarians would also provide valuable insight into communication on this topic. Fourth, although outside the scope of this study, a logical expansion would be to explore approaches used by behaviourists to address or resolve fear/anxiety in their clients’ horses. Finally, a more detailed exploration regarding the educational props and exercises different equine behaviour consultants use in behaviour consultations would be valuable to share within the sector. 

This study is the first of its kind to conduct detailed interviews with equine behaviourists and to synthesise the results to provide information that will be useful to behaviourist and animal welfare professions. It will enable an increased understanding of the experiences of others and provide valuable information for students and early career professionals.

## 5. Conclusions

The study provides an informed insight into the recognition of fear and/or anxiety in horses by horse caregivers as perceived by equine behaviourists. The qualitative research has resulted in the documentation of a range of perspectives and experiences that can be shared between professionals in this field. The key themes that emerged throughout several of the questions were: that fear and/or anxiety is very poorly recognized by horse caregivers; that the more overt signs of more extreme fear are more likely to be recognised than the more subtle signs; and that fear and/or anxiety is often misinterpreted by horse caregivers. When considering which explanations work best to help caregivers to understand that their horse is fearful or anxious, three themes emerged: explanations that focused on drawing parallels with human experiences of fear; explanations focusing on the ethological elements of fear; and using guided observations of equine behaviour to highlight subtle behavioural signs of fear and/or anxiety. The interviewees had mixed feelings regarding whether there are positive changes for horses and attributed a lack of time, or perceived lack of time, as a key barrier to significant sustainable change. The study provides an informed insight into the recognition of fear and anxiety of horses by their owners in the United Kingdom as well as tried and tested approaches to conversations to change this. The collaborative synthesis of experience and techniques across the equine behaviour sector, together with the information gained regarding perception of equine caregivers, could be a valuable approach to improve the effectiveness of behaviour consultations and welfare initiatives.

## Figures and Tables

**Table 1 animals-12-02904-t001:** The eight questions and the resulting summary and themes.

Question	Themes
How well do people recognise fear and/or anxiety in horses?	Theme 1: Caregivers are extremely poor at recognizing fear and anxiety in horses. Theme 2: Some caregivers do recognise behavioural signs indicating fear and/or anxiety but only the big signs (e.g., rearing, running away) rather than the small signs (e.g., tension in face, subtle avoidance behaviours such as a hesitant gait). Theme 3: fear and/or anxiety behaviour is often misinterpreted or mislabelled.
How do caregivers tend to respond when told their horses are fearful and/or anxious?	Theme 1: Some clients take the information that their horses are fearful/anxious on board. Theme 2: Those clients who are sceptical of fear/anxiety being a factor and continue to label the cause of the behaviour as something else (e.g., horse is being lazy or naughty). This group fits into theme 3 of question 1.
Is fear and/or anxiety something you tend to always talk about in a consult or only if it is their primary reason for calling you out?	Theme 1: Some interviewees said ‘always’ (six out of nine people) Theme 2: Interviewees who would be more cautious (three out of 9 people.
What explanations or analogies have worked best in getting across to people that their horses are fearful and/or anxious?	Theme 1: Likening their horse’s fear/anxiety to human experiences of fear/anxiety. Theme 2: Framing explanations from the angle of ethology. Theme 3: Focussing on observations of the horse in front of them.
Do you come across resistance? People not believing their horses are fearful and/or anxious?	Theme 1: The importance of encouraging the owners to reflect on their own behaviour and understanding of fear in horses and humans alike. Theme 2: Reflections regarding the interpersonal skills needed to be an effective behaviourist.
Do you find there is a difference in how people perceive fear and/or anxiety if the cause of it is a thing (e.g., hose, car) or if the cause is the owner themselves (e.g., training method)?	Theme 1: People are more compassionate to the horse when they are fearful of an object than the behaviour of a person themselves. Theme 2: Questioning whether the emotional attachment is as positive from a horse’s perspective as it is from the caregiver’s.
Are things changing?	This question generated a spectrum of responses from positive to negative. Often as interviewees continued to talk, they tended to go from one end of the spectrum to the other, irrespective of which end they started at.
Is there any other information you’d like to share about your experiences with your client’s perception and understanding of fear and/or anxiety in horses?	Theme 1: The challenge of client’s perceptions regarding the time they are prepared to invest in addressing their horse’s behaviour. Theme 2: The barrier regarding that fear and/or anxiety behaviors in horses are so ‘normed’ in equestrian society that it can be difficult to go against that ‘norm’.

## Data Availability

Anonymised transcripts of interviews are available from the authors.

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
