# Peer review of "Perceptions of Fear and Anxiety in Horses as Reported in Interviews with Equine Behaviourists"

_animals, 2022, doi:10.3390/ani12212904_

Round 1

Reviewer 1 Report

Thank you for a great concept and a nice article. Please consider the following edits.

Line 23-27: This is a complete copy of what is written in a simple summary. It would be great if you could rewrite this.

Line 30-40: Again, this is a simple copy/paste of what is written in the summary. It would be great if you could rewrite the abstract.

Line 110-111: Please explain to the reader what the very small sample size might mean and how it might affect the results of this study.

Line 113: What are the possible confounding factors? It would be appropriate to state and cite them here.

Line 143: I would advise against pasting the participant's response. Please present your results in a more direct and clear manner. What are the results of your study? Not the sentences told by the participants.

Line 598-600:  Please discuss how caregivers' failure to recognize fear and anxiety-related behaviors affects horses’ overall behavior and welfare.

Line 584-630: Your discussion section is far too brief. I would recommend you to discuss all of your results and consider doing it question (1-8 variable) wise.

Author Response

Reviewer 1:

Thank you for a great concept and a nice article. Please consider the following edits.

We thank Reviewer 1 for their comments, support and the time taken to review this paper.

Line 23-27: This is a complete copy of what is written in a simple summary. It would be great if you could rewrite this.

We have edited the simple summary and abstract so there is less overlap between the text of these two sections.

Line 30-40: Again, this is a simple copy/paste of what is written in the summary. It would be great if you could rewrite the abstract.

We have edited the simple summary and abstract so there is less overlap between the text of these two sections.

Line 110-111: Please explain to the reader what the very small sample size might mean and how it might affect the results of this study.

We have added to the end of this section – “The small sample size is not a concern for this study as it is a qualitative study and there were high levels of agreement between participants. Also, considering the number of practicing active behaviourists in the United Kingdom, the sample size is fairly proportionate. This study focusses on what the participants said, based on their extensive experience with clients, rather than how many participants gave certain responses.”

Line 113: What are the possible confounding factors? It would be appropriate to state and cite them here.

We have added this example: For example, a prerequisite for membership of ABTC is that behaviourists use evidence-based approaches to behaviour modification whereas some practicing behaviourists without ABTC membership have not studied learning theory, ethology, or the physiology of stress and use methods that compromise animal welfare.

Line 143: I would advise against pasting the participant's response. Please present your results in a more direct and clear manner. What are the results of your study? Not the sentences told by the participants.

The Editor recognises that this style of citing quotes is appropriate for a qualitative study such as this and advised against actioning this point. The other reviewers also supported the current format.

Line 598-600:  Please discuss how caregivers' failure to recognize fear and anxiety-related behaviors affects horses’ overall behavior and welfare.

We have added “This affects horse welfare because if caregivers do not identify when horses are fearful, anxious or ‘stressed’ they are likely to continue interacting, handling, and training horses in a way that causes those negative affective states, and consequently cause behaviours that could be dangerous as well as compromise horse welfare.”

Line 584-630: Your discussion section is far too brief. I would recommend you to discuss all of your results and consider doing it question (1-8 variable) wise.

Due to the way the results are written we have not restructured the discussion question by question but have added details and discussion points to the discussion section, according to suggestions from the other referees and Editor.

Reviewer 2 Report

The recognition of the emotional state of animals is a research field that has been growing in the past decade, mainly because it can be used as a part of a comprehensive welfare assessment. This study shows in detail how equine behaviorists recognize pain, and how this recognition can be perceived differently by the owner, particularly when they are not trained to recognize changes in facial expression or body language. This article reflects these factors that influence the lack of recognition and the search for training that allows demonstrating the need for precise indicators of the species in equines. I have left some comments hoping they can help the authors improve their manuscript.

Line 25: Please, include the objective of the present study.

Line 48-49: I suggest briefly including in which instances the “equine face” has been used since subtle changes in the facial expression are also mentioned in the present study https://pubmed.ncbi.nlm.nih.gov/35017848/

Lines 57-60: Could the authors include one or two examples that can trigger fear and stress in horses? In this way, the reader can associate routine or common practices as a challenge to the welfare of horses.  

Line 82: Please, write the name of the author of reference 15 before citing their study. The same applies in line 593. Revise the citation style of the journal.

Line 97: Consider writing the objective of the study before stating the hypothesis.

Line 110: Describe how the sample size was calculated.

Line 264: Revise if “anx” needs to be changed to “anxiety”

Lines 589-593: I suggest discussing after this sentence that one of the challenges in recognizing fear or anxiety in animals such as a horse is that their prey status almost obligates them to conceal any factor that could make them vulnerable to predators. Therefore, equines might have developed more subtle signs of negative emotions. The authors could refer to this study:  10.1016/j.cub.2013.07.070

 Lines 597-600: This is an important factor that needs to be expanded a little bit. When mentioning animal body language or facial expression related to emotion (whether negative or positive), the lack of experience or knowledge of the natural behavioral repertoire of animals is one factor that causes differences even in experimental designs where a researcher has more experience than others identifying a position change of the ears or eye closing. It would be relevant to include a sentence mentioning this.

Likewise, the consequences of anthropomorphism are not only present in small animals, but in the horse as well. Some of the responses of the interviewees give a hint to that. Therefore, I suggest including a couple of lines about how anthropomorphism influences the perception of animal emotions (negatively and positively). This article may help the authors: https://doi.org/10.3390/ani11113263

Line 601-610: As a part f the limitations of the study, the authors could include, together with the experience of the owners, their sex and age. Some studies related to pain recognition have shown that women and men tend to rat pain differently when assessed through behavioral aspects.

Author Response

Reviewer 2:

We thank Reviewer 2 for their comments, support and the time taken to review this paper. We especially appreciate the suggested references to include.

The recognition of the emotional state of animals is a research field that has been growing in the past decade, mainly because it can be used as a part of a comprehensive welfare assessment. This study shows in detail how equine behaviorists recognize pain, and how this recognition can be perceived differently by the owner, particularly when they are not trained to recognize changes in facial expression or body language. This article reflects these factors that influence the lack of recognition and the search for training that allows demonstrating the need for precise indicators of the species in equines. I have left some comments hoping they can help the authors improve their manuscript.

Line 25: Please, include the objective of the present study.

We added the objective.

Line 48-49: I suggest briefly including in which instances the “equine face” has been used since subtle changes in the facial expression are also mentioned in the present study https://pubmed.ncbi.nlm.nih.gov/35017848/

We added: In particular, the equine pain face has been researched in the context of castration surgery [4], laminitis [5] and pain induced by tourniquet on the front leg and capsaicin applied topically [6]. The inclusion of behavioural indicators alongside physiological markers is recognised as an important factor in the recognition of pain [7]).

Lines 57-60: Could the authors include one or two examples that can trigger fear and stress in horses? In this way, the reader can associate routine or common practices as a challenge to the welfare of horses.  

Have added (after the Rioja-Lang citation): Example triggers of such stress include pain, changes of home and/or ownership and inappropriate riding and 'use’”.

Line 82: Please, write the name of the author of reference 15 before citing their study. The same applies in line 593. Revise the citation style of the journal. Done

Line 97: Consider writing the objective of the study before stating the hypothesis. Done

Line 110: Describe how the sample size was calculated.

Further information about sample size has been added elsewhere as this was also mentioned by another referee.

Line 264: Revise if “anx” needs to be changed to “anxiety” Done

Lines 589-593: I suggest discussing after this sentence that one of the challenges in recognizing fear or anxiety in animals such as a horse is that their prey status almost obligates them to conceal any factor that could make them vulnerable to predators. Therefore, equines might have developed more subtle signs of negative emotions. The authors could refer to this study:  10.1016/j.cub.2013.07.070

We have added: Although horses are prey animals and typically display subtle signs of pain, fear and stress [added reference provided, thank you], the consistency with which behaviourists recognise these subtle signs suggests that the subject can and should be a key part of equestrian education.

Lines 597-600: This is an important factor that needs to be expanded a little bit. When mentioning animal body language or facial expression related to emotion (whether negative or positive), the lack of experience or knowledge of the natural behavioral repertoire of animals is one factor that causes differences even in experimental designs where a researcher has more experience than others identifying a position change of the ears or eye closing. It would be relevant to include a sentence mentioning this.

We appreciate this point, and this was the reason that the participants were recruited from ABTC-registered behaviourists – because they meet at least a standard regarding ability to recognise fear and anxiety in horses and have a strong knowledge of the equine ethogram. We elaborated on this in the methodology section.

Likewise, the consequences of anthropomorphism are not only present in small animals, but in the horse as well. Some of the responses of the interviewees give a hint to that. Therefore, I suggest including a couple of lines about how anthropomorphism influences the perception of animal emotions (negatively and positively). This article may help the authors: https://doi.org/10.3390/ani11113263

Thank you for highlighting this discussion point. We have added “Anthropomorphism influences the perception of animal emotions both negatively and positively [30,31]. This is reflected in the participants’ responses not only regarding anthropomorphism used by their clients but also by the participants themselves. Behaviourists are skilled in using ‘constructive  anthropomorphism’ [32] to help their clients understand the cause of their horses behaviours and  to encourage their clients to develop empathic responses. Discarding the potential that horses feel these emotions is likely to reduce such understanding and empathy, to allow the maintenance of client behaviour that causes fear and ultimately to reduce animal welfare.”

Line 601-610: As a part of the limitations of the study, the authors could include, together with the experience of the owners, their sex and age. Some studies related to pain recognition have shown that women and men tend to rate pain differently when assessed through behavioral aspects.

Although we appreciate the reasoning behind this suggestion, we do not think it appropriate to mention the sex and age of behaviourists due to the study being qualitative (there are no useful stats that could be done on this sample size) and it would significantly reduce anonymity.

Reviewer 3 Report

This is a well-written paper outlining horse caregiver perceptions of fear and anxiety in their horses as understood by equine behaviourists. The key finding that most caregivers do not recognize behavioural signs of fear and anxiety is not surprising but is important to help educate and effect human behaviour change in address these issues. My only comments are grammatical in nature. Well done!

L91 – add “in horses” before “…is perceived by clients”

L100 – please write out ABTC in full with first time use of the acronym (this is done in the abstracts but not in the body of the paper)

L120 – might be helpful to clearly indicate that you contacted 28 behaviour consultants (if that is indeed the number on the list)

L127 – add the word “to” before “37 minutes…”

L131 – clients, not clients’

L137 – delete one of the “online” words

Table 1: Questioning spelled incorrectly in theme 2 of Q6

Author Response

Reviewer 3:

We thank Reviewer 3 for their comments, support and the time taken to review this paper. We have made all the suggested changes to correct the grammatical errors.

This is a well-written paper outlining horse caregiver perceptions of fear and anxiety in their horses as understood by equine behaviourists. The key finding that most caregivers do not recognize behavioural signs of fear and anxiety is not surprising but is important to help educate and effect human behaviour change in address these issues. My only comments are grammatical in nature. Well done!

L91 – add “in horses” before “…is perceived by clients”

L100 – please write out ABTC in full with first time use of the acronym (this is done in the abstracts but not in the body of the paper)

L120 – might be helpful to clearly indicate that you contacted 28 behaviour consultants (if that is indeed the number on the list)

L127 – add the word “to” before “37 minutes…”

L131 – clients, not clients’

L137 – delete one of the “online” words

Table 1: Questioning spelled incorrectly in theme 2 of Q6

Round 2

Reviewer 2 Report

The authors have made each of the suggested changes.

The manuscript is now more complete and comprehensive.

I suggest for it to be published.

Author Response

Thank you for your comments, we are very appreciative.